# Optimizing Confined Nitride Trap Layers for Improved Z-Interference in 3D NAND Flash Memory

Yeeun Kim [ID], Seul Ki Hong [ID] and Jong Kyung Park *

Department of Semiconductor Engineering, Seoul National University of Science & Technology, Seoul 01811, Republic of Korea; yeeun0713@seoultech.ac.kr (Y.K.); skhong@seoultech.ac.kr (S.K.H.)
* Correspondence: jkpark1@seoultech.ac.kr

**Abstract:** This paper presents an innovative approach to alleviate Z-interference in 3D NAND flash memory by proposing an optimized confined nitride trap layer structure. Z-interference poses a significant challenge in 3D NAND flash memory, especially with the reduction in cell spacing to accommodate an increased number of vertically stacked 3D NAND flash memories. While the confined nitride trap layer device designed for complete isolation of the trapping layer in three dimensions effectively reduces Z-interference, the results showed substantial variations based on the confined structure. To clarify this issue, we compared three distinct confined nitride trap layer structures and investigated their impact on Z-interference. Our findings indicate that the rectangle structure exhibited the most significant mitigation, implying that differences in the electric field applied to the poly silicon channel, which is influenced by the structure, and the increase in effective channel length are effective strategies for alleviating Z-interference. The proposed structure undergoes a comprehensive examination through technology computer-aided design (TCAD) simulations. Additionally, we introduce a practical process flow designed to minimize Z-interference.

**Keywords:** 3D NAND flash memory; confined nitride trap; Z-interference

## 1. Introduction

Due to the rapid increase in data, there is a growing demand for high-capacity memory [1–3]. Consequently, the structure of NAND flash has transitioned from a 2D planar configuration to a 3D vertical configuration [4–6]. To increase the bit density in 3D NAND flash, it is necessary to stack multiple layers [7,8]. However, this has led to challenges in etching processes, becoming progressively difficult [9]. In response to these challenges, vertical scaling has become essential. As the spacing between word lines (WLs) decreases, issues regarding the reliability of the device arise. One such issue stems from WL–WL interference, known as Z-interference. Z-interference occurs when an attack cell is programmed, altering the threshold voltage (Vth) of the victim cell, thereby affecting the distribution window of the Vth and reducing the margin between distributions [10,11]. The main cause of Z-interference arises from electrons programmed within the attack cell, leading to a reduction in the electron density of the poly-Si channel beneath the victim cell. Consequently, compensating for this electron density reduction necessitates applying a higher read voltage to the victim cell, consequently elevating its Vth. To mitigate Z-interference effectively, it becomes imperative to minimize the change in electron density within the poly-Si channel at the bottom of the victim cell caused by electrons programmed within the attack cell. Various research endeavors have been pursued to address this issue, primarily focusing on enhancing operational conditions. Initially, a method proposing the application of a high read voltage to the adjacent WL during read operations was introduced [12,13]. However, this approach poses the risk of exacerbating read disturbance across the entire block due to the elevated read bias of the adjacent WL. As an alternative, a method advocating for the reduction of the pass voltage of the adjacent WL during program operations

was proposed [14]. This method facilitates precise control over the electron distribution within the charge trap layer programmed within the attack cell, thereby diminishing the changes in electron concentration within the lower channel of the victim cell. Nevertheless, this approach compromises the channel-boosting characteristics of the selected WL during program operations, consequently exacerbating program disturbance. Secondly, structural modifications of the device have been proposed. One such method involves initially forming a channel hole and subsequently transforming its structure from concave to convex [9]. However, this method introduces the challenge of deteriorating data retention characteristics due to increased electron concentration within the central area of the cell in the convex structure. More recently, to address Z-interference in 3D NAND flash, a confined nitride trap layer device has been proposed, offering complete three-dimensional isolation and obstructing lateral migration paths [15]. Despite the considerable complexity associated with implementing this method, it presents a promising solution for simultaneously alleviating the trade-off problems of Z-interference and retention characteristics of the convex structure, emerging as a crucial technology for continuous vertical scaling. Nevertheless, there remains a dearth of comprehensive research elucidating the optimal characteristics of the confined nitride structure with regard to Z-interference. Therefore, we employed technology computer-aided design (TCAD) simulations to investigate Z-interference based on the confined nitride trap layer structure in 3D NAND flash. By respectively modifying the structure of the confined nitride trap layer in the attack and victim cells, the root cause of the change, which resulted in the Z-interference, was systematically analyzed. Consequently, we propose the structure of a confined trapping layer with the most improved Z-interference and suggest a process flow to achieve such a structure.

## 2. Simulation Set Up

Figure 1a,b show transmission electron microscopy (TEM) images of cross-sections of a 3D NAND with a confined nitride trapping layer [13]. Figure 1c,d show schematic cross-sections of the 3D NAND with the confined nitride trapping layer used in the TCAD simulation. In 3D NAND structure, there are five cells, along with string-selection transistors, such as the drain select line (DSL) and source select line (SSL), which can selectively connect a cell string to bit line (BL)/source line (SL). Tungsten was used as the gate material. The gate length is 25 nm, space length is 20 nm and channel thickness is 7 nm. The thickness of the blocking oxide/nitride/tunneling oxide layers is 7 nm, 5.5 nm, and 4.5 nm, respectively. For the simulation of the confined nitride trap layer, three different structures were established, as illustrated in Figure 2a–c. Upon examining the cross-section of the 3D simulation depicted in Figure 2a–c, the area of the charge trap nitride in the chamfer 1 and chamfer 2 structures was set to be identical. Specifically, among the lengths of the two parallel bases of the trapezoid in cross-section, the length of the longer base was fixed at 20 nm, matching the WL length, while the length of the shorter base was set to 10 nm. Conversely, in the rectangular structure, a charge trap nitride with a rectangular cross-section was employed with both bases set to 20 nm. Additionally, since the 3D simulation was conducted using the cylindrical function, it can be inferred that the total volume of the charge trap is considerably larger in the rectangular structure compared to chamfer 1 and chamfer 2, resulting in a higher trap density as well. The boron doping concentration of the poly-silicon channel is $1 \times 10^{15} \mathrm{cm}^{-3}$, and the phosphorus source/drain doping concentration is $1 \times 10^{19} \mathrm{cm}^{-3}$. Program/erase/read voltage conditions are described in Table 1. WL2 was designated as the victim cell, with the adjacent cell WL1 programmed as the attack cell, and the extent of Z-interference was determined by examining the shift in the Vth of WL2. While assessing Z-interference, both the initial and the post-programmed Vth of the attack cell are crucial factors, alongside the degree to which the victim cell is programmed. Even if the programmed level of the attack cell remains constant, Z-interference diminishes as the Vth of the victim cell increases [16]. Hence, in the simulation, an incremental step pulse erase (ISPE) operation was conducted prior to programming to establish identical erased Vth values across all split structures. Subsequently, instead of employing the same

program voltage during program operation, the program voltage was tailored for each structure to achieve consistent post-programmed Vth in each split structure. Initially, the programmed Vth of the attack cell was confirmed through incremental step pulse program (ISPP) operation. Following this, for the sake of simulation convenience, a single pulse was applied to proceed with the program operation. Throughout programming, a fixed voltage of 9 V was applied not only to the adjacent WL but also to the entire WL. Meanwhile, the read voltage was set at 5.5 V, with 3.3 V applied to SSL/GSL and 0.5 V to BL. During this process, the bias of the selected cell, victim WL, was swept from −5 V to 5 V to assess the cell state, with the voltage at which the current flowing through the entire string reached 50 nA established as the Vth. Even after completing the program operation of the attack cell, the bias of victim WL was similarly swept from −5 V to 5 V, and the voltage at which the current was 50 nA was designated as the Vth. Subsequently, the threshold voltage before program operation and the degree of change in Vth after program operation were defined as Z-interference. In a subsequent analysis, identical Vth values were applied to the victim cells before and after the program to determine the change in electron density within the poly-Si channel after the program. In TCAD device simulations, the Shockley–Read–Hall (SRH) model, Poole–Frenkel model, and non-local tunneling model were employed. The gate-all-around (GAA) 3D NAND cell was modeled using the cylindrical function. Furthermore, the physics model for polysilicon utilized a doping-dependent model and a high-field saturation model for mobility. Additionally, studies have indicated that Z-interference can be affected by grain boundary traps in polysilicon channels [10]. Nevertheless, in this simulation, to simplify matters, this effect was excluded, and only the influence of structural modifications in the cell was taken into account.

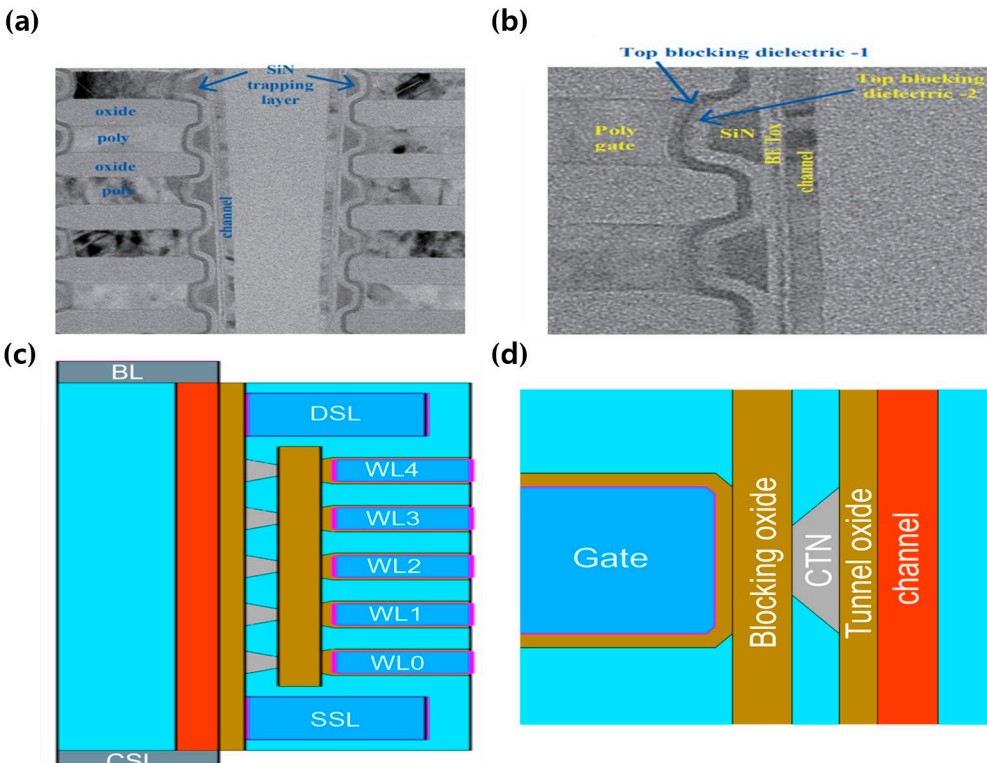

**Figure 1.** (**a**,**b**) TEM image of the cross-section of 3D NAND with a confined nitride trapping layer. Reprinted/adapted with permission from Ref. [15], 2019, VLSI. (**c**) Schematic cross-section of 3D NAND with a confined nitride trapping layer used in the simulation. (**d**) Detailed zoom-in view of (**c**).

**Table 1.** Simulation voltage condition.

|  | Program | Erase | Read |
|---|---|---|---|
| Selected cell | 18.5 V~19 V | 0 V | −3 V~5 V |
| Unselected cell | 9 V | 0 V | 5.5 V |
| BL | 0 V | 22 V | 0.5 V |
| DSL | 3.3 V | floating | 3.3 V |
| SSL | 0 V | floating | 3.3 V |
| SL | 2 V | 22 V | 0 V |
| Time | 20 µs | 0.5 ms | - |

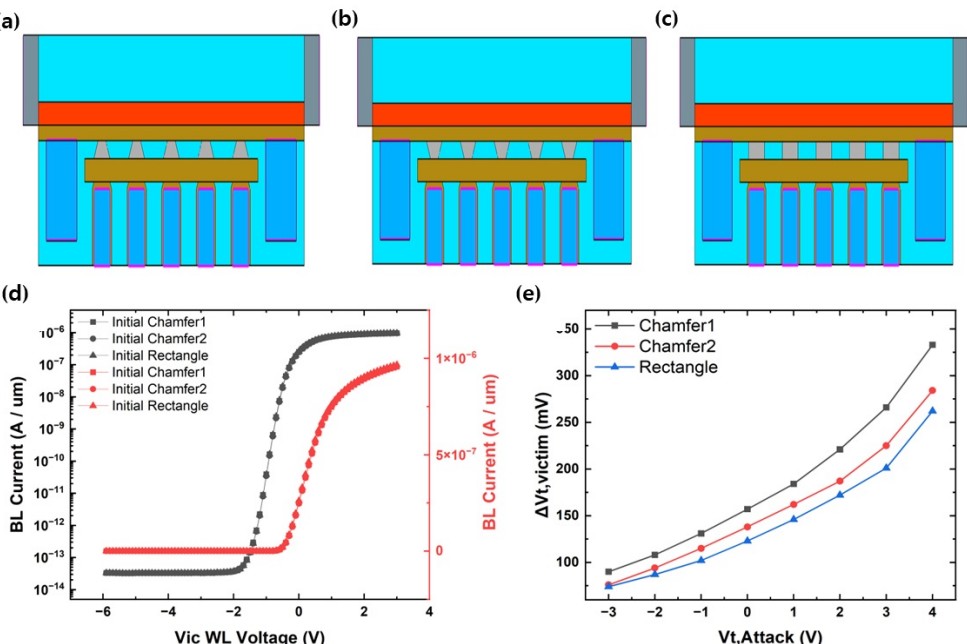

**Figure 2.** Simulated structure: (**a**) chamfer 1, (**b**) chamfer 2, and (**c**) rectangle. (**d**) $I_D$-Vg characteristic curves. (**e**) Z-interference simulation results.

## 3. Investigation of Z-Interference

In this study, Z-interference simulations were conducted using a 3D NAND with three different confined nitride trap layer structures. Figure 2a–c show the 3D NAND devices specified as chamfer 1, chamfer 2, and rectangle, respectively, according to the structure of the confined nitride trap layer. Figure 2d shows the initial $I_D$-$V_G$ characteristics for the three structures. The differences in on current, off current, and subthreshold swing characteristics are all negligible across the three structures. Figure 2e shows the Z-influence of the confined nitride trap layer structure. The simulations were performed by erasing all cells to the same Vth before programming the attack cell. Z-interference is significantly observed in the order of chamfer 1, chamfer 2, and rectangle.

To analyze the effect of the confined nitride trap layer structure on Z-interference, simulations were firstly conducted by changing only the confined nitride trap layer structure of the attack cell. Figure 3a–c show the electron trapped charge in a 3D NAND device after programming the attack cell. It can be observed that the patterns of trapped electron appear differently depending on the charge trap layer structure. Figure 3d shows simulation results of Z-interference in the three different structures. When only the confined nitride trap layer structure of the attack cell is changed, unlike when the trap layer structure of all cells is changed, Z-interference is largely evaluated in the order of rectangle, chamfer 1, and chamfer 2. To analyze this cause, we confirmed the profile of the electron trapped charge programmed in the attack cell. Generally, the reason for the change in Z-interference is that electrons trapped in the trap layer create an electric field in the direction of the poly-Si

channel, subsequently altering the electron concentration of the poly-Si channel at a given read voltage [10]. Therefore, the charge trapped in the area closest to the tunnel oxide is expected to have the most significant impact on the conduction of poly-Si, and the electron trapped charge in this area was verified. As depicted in Figure 3d, the results indicate that the rectangle structure exhibits the widest distribution of electrons compared to chamfer 1 and chamfer 2. Through this observation, it can be inferred that the rectangle structure exerts the most substantial influence on the electron concentration distribution of the victim cell, thereby deteriorating the Z-interference characteristics the most. Meanwhile, when comparing chamfer 1 and chamfer 2, it is confirmed that the profile of trapped electrons in chamfer 1 is much narrower than that in chamfer 2. Nevertheless, the Z-interference phenomenon can be observed to be worse in chamfer 1 than in chamfer 2. It is unlikely that this discrepancy is solely attributable to an area effect between chamfer 1 and chamfer 2, as chambers 1 and 2 were designed with distinct shapes yet identical areas during the experiment. Consequently, we encountered difficulty in explaining all the minor variations in Figure 3 based solely on area considerations. Moreover, while there is no significant difference in Z-interference between chamfer 1 and chamfer 2 when the Vth of the attack cell is 1.5 V or less, a notable discrepancy in Z-interference emerges when the Vth of the attack cell exceeds 1.5 V. Consequently, it is challenging to attribute these differences solely to area effects.

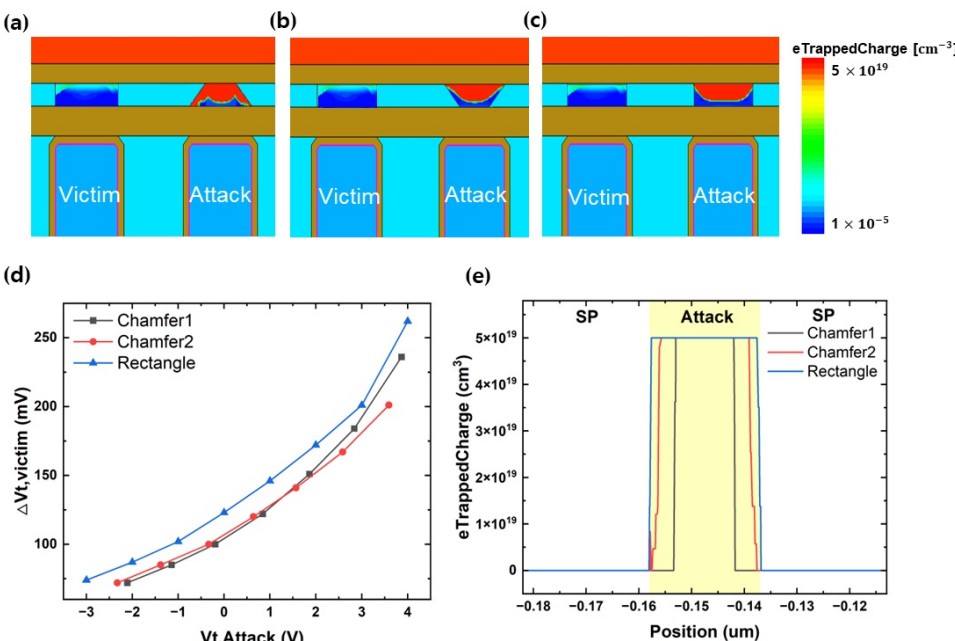

**Figure 3.** Electron trapping results in the charge trap nitride (CTN) layer within a simulated structure with attack cell altered. (**a**) Chamfer 1. (**b**) Chamfer 2. (**c**) Rectangle. (**d**) Z-interference simulation results from (**a**–**c**,**e**). Location of electron charge trapped at the TOX/CTN (charge trap nitride) interface.

To further analyze the cause of this discrepancy, the electron density in the poly-silicon channel before and after programming was compared in Figure 4a. As expected, there is no difference in the electron density of the poly-Si channel under the victim cell before programming. However, after the attack cell is programmed, it is observed that the channel electron density of chamfer 1 drops more significantly. Moreover, it was confirmed that the electron density of the poly-Si channel at the space oxide area between the attack cell and the victim cell also decreased. To confirm this difference in electron density, the electric field near the attack cell was verified in Figure 4b,c when the read voltage was applied immediately after programming the attack cell. Looking at the results, it can be seen that the vector direction of the electric field due to the read voltage is more concentrated toward the bottom of the attack cell in the chamfer 1 structure than in the chamfer 2 structure. Generally, when a read voltage is applied, a fringing electric field must be applied to the

channel in the space area between the attack cell and the victim cell to create a path for electrons to conduct [17]. However, in the chamfer 1 structure, the electric field of the attack cell is rather concentrated at the bottom of the attack cell, causing a significant drop in electron concentration in the space region. Therefore, this means that a larger voltage must be applied to the victim cell during a read operation to allow the same cell current to flow, and it can be assumed that the Z-interference phenomenon in chamfer 1 is deteriorated compared to chamfer 2 due to this feature. Due to this phenomenon, upon examining Figure 3d, it is evident that there is no significant difference in Z-interference between chambers 1 and 2 until the program Vt of the attack cell reaches approximately 1 V. However, it can be inferred that as the program level exceeds 1 V, the difference in Z-interference becomes more pronounced due to the dispersion effect of the electric field caused by the electrons in the charge trap layer. Analyzing the results allows for an explanation of the Z-interference on the victim cell when only the structure of the attack cell is modified. However, fully interpreting the extent of influence on Z-interference remains challenging when all confined trap layers of the entire string are altered.

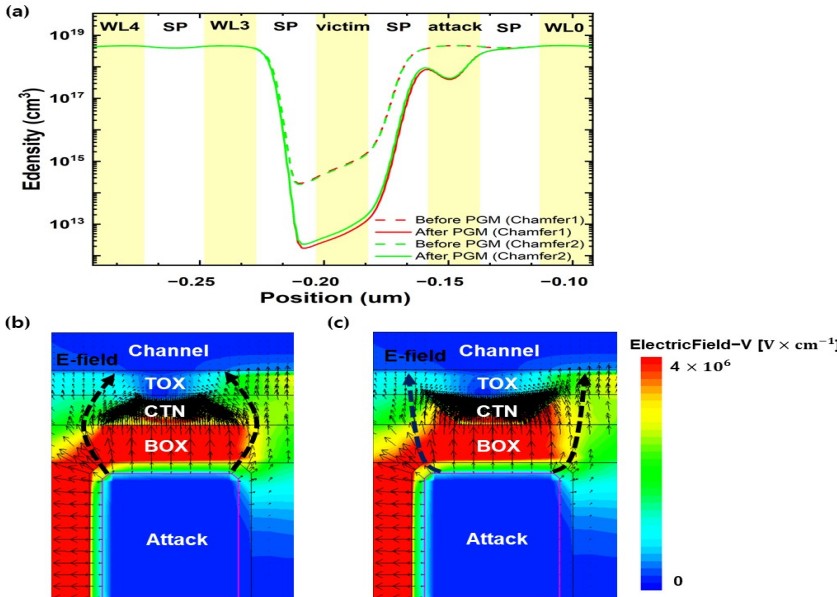

**Figure 4.** (**a**) Electron density profiles for chamfer 1 and chamfer 2 with the structure of the attack cell altered. In this situation, victim WL bias is at the threshold voltage (Vth), and the other WLs are at the read voltage. (**b**,**c**) Electric field for chamfer 1 and chamfer 2. In this situation, victim WL bias is at Vt, and the other WLs are at the read voltage.

Therefore, simulations were conducted by altering only the confined trap layer structure of the victim cell, as illustrated in Figure 5a–c. Figure 5d presents the results of Z-interference experiments for these three structures. When only the confined trap layer structure of the victim cell is modified, Z-interference appears larger in the following order: chamfer 1, chamfer 2, and rectangle. This result contrasts with the Z-interference result when only the confined trap layer structure of the attack cell was changed. Consequently, as depicted in Figure 2e, it matches well with the Z-interference result when all confined trap layers of the entire string are altered.

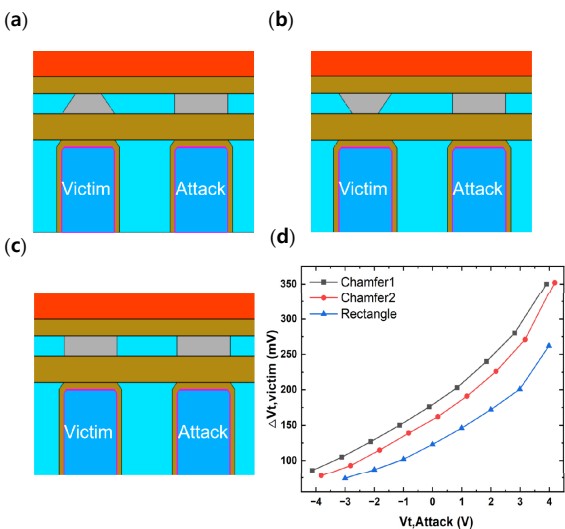

**Figure 5.** Simulated structure with victim cell structure altered. (**a**) Chamfer 1. (**b**) Chamfer 2. (**c**) Rectangle. (**d**) Z-interference simulation results from (**a**–**c**).

To analyze this phenomenon, the energy band profile of the poly-Si channel before and after programming the attack cell in the three structures was examined in Figure 6a. During the read operation, both before and after the attack cell's program, a fixed voltage near the Vth is applied to the victim cell, and a read voltage is applied to the unselected cell. The distance between the energy barriers in the bit line direction at a point approximately 0.2 eV higher than the energy level of the source line was defined as the effective gate length in the victim cell. Observing the results, before the attack cell program, the effective gate length of the rectangle structure appears to be the largest. This can be attributed to the fact that, as demonstrated in the results in Figure 3e, the charge profile after programming is the widest in the rectangle structure. In other words, this programmed charge provides a screening effect towards the poly-Si channel when the same read voltage is applied to the victim cell during read operation, leading to the widest effective channel length. Additionally, when examining the effective gate length after programming, the rectangle structure exhibits the least amount of change. This suggests that the effective gate length of the victim cell in the rectangle structure is the widest before the attack cell is programmed, providing the greatest immunity to the short-channel effect phenomenon and the best gate controllability. Consequently, due to excellent gate controllability, even after the attack cell is programmed, the impact the attack cell on the victim cell is minimal, and it is inferred that the amount of change in the effective gate length is the smallest. As a result, the conduction band difference among the three structures before and after the attack cell is programmed appears large in the following order: chamfer 1, chamfer 2, and rectangle. Figure 6b displays the conduction band energy in the three structures before programming the attack cell. It precisely matches with the trend of Z-interference in Figure 5d. This analysis indicates that maximizing the effective gate length before the attack cell's program enhances the gate controllability of the victim cell. Therefore, securing a larger effective gate length is likely to minimize Z-interference during the attack cell's program operation. Summarizing the aforementioned results, when the attack cell in Figure 3 is altered to a rectangular structure, the profile of trapped electron charge widens, resulting in the most pronounced deterioration of Z-interference. However, it is evident that changing the victim cell's structure to a rectangular shape effectively counteracts the degradation induced by the attack cell, showcasing optimal Z-interference characteristics. Hence, it can be concluded that enhancing the gate controllability of the victim cell, rather than focusing solely on the intensity of the attack, stands out as one of the critical aspects for improving Z-interference.

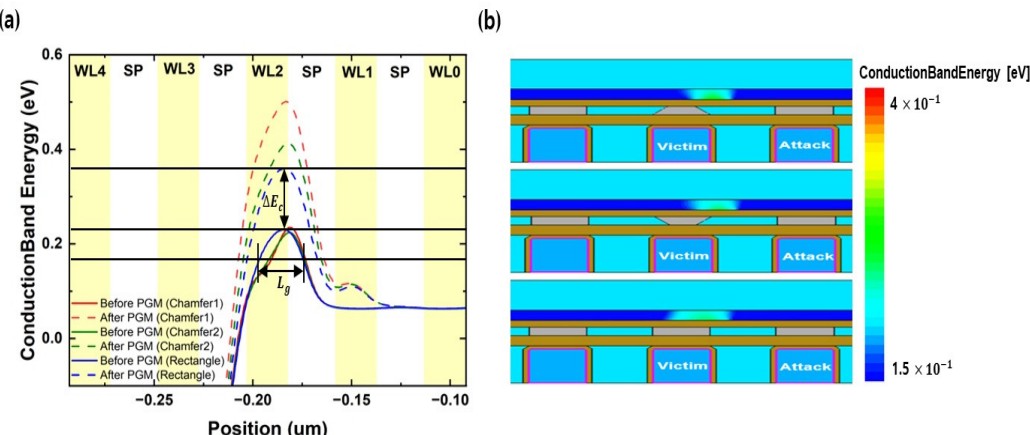

**Figure 6.** Conduction band profiles of chamfer 1, chamfer 2, and rectangle with the structure of the victim cell altered. In this situation, victim WL bias is at Vt, and the other WLs are at the read voltage. (**a**) Comparison of the conduction band energy before and after programming the attack cell. (**b**) Conduction band profiles of chamfer 1, chamfer 2, and rectangle before programming the attack cell.

## 4. Process Flow

Figure 7a–e illustrate the general process flow of creating a confined trap layer structure in 3D NAND flash [15]. Initially, alternating layers of oxide and nitride are deposited on silicon. Subsequently, sidewall lateral etching is performed exclusively on the nitride layer. Achieving high selectivity and uniformity across all layers is crucial for this process. However, Figure 7b reveals that the etching of nitride at the oxide/nitride interface progresses more slowly than in the middle part of nitride. This suggests that the process may result in a nitride trap layer structure, such as the chamfer 2 structure. In this structure, blocking oxide and silicon nitride trapping layers are deposited, as depicted in Figure 7c. At this stage, the silicon nitride trapping layer is deposited as thick as 20 nm. In this way, the nitride film deposited inside the plug hole fills the entire recessed region and flattens the entire inner surface of the plug. Therefore, by repeatedly performing the dry etching and wet etching process, a confined trapping layer can be formed in a self-aligned manner [18]. This charge trap layer pullback process separates the trapping layer in the Z direction, followed by the deposition of tunneling oxide. Thin poly channel deposition follows, and the word line is formed using tungsten metal by replacing the stacked nitride films. To address these challenges, Figure 7f–j present a proposed process for manufacturing 3D NAND flash with a rectangle structure that minimizes Z-interference. Initially, a layer consisting of alternating oxide and two nitride layers is deposited on silicon. Specifically, nitrides in the regions adjacent to the oxide exhibit higher etch rates compared to nitrides in the intermediate regions. Generally, when depositing silicon nitride material using chemical vapor deposition (CVD), it is reported that the etch rate can be controlled more effectively by increasing the $SiH_4$ gate flow rate and reducing the $NH_3$ and $N_2$ gas flow rates [19,20]. Therefore, based on this process method, the following procedure can be implemented as an example to create a rectangular structure within the chamfer 2 structure. In other words, the relative ratio of the $SiH_4$ gas flow rate is decreased to enhance the etching rate in the area adjacent to the oxide, while the $SiH_4$ gas flow rate is increased to reduce the etching rate in the bulk area farther from the oxide interface. This approach enables subsequent sidewall etching of the two nitride layers to resolve the issue of slow etching of the nitride at the interface with the oxide, as demonstrated in Figure 7g, due to the faster etching rate of the region adjacent to the oxide. The process then proceeds similarly to Figure 7c–e. By utilizing silicon nitride materials with different compositions in a multilayer stack of oxide and nitride, it is believed that these materials will not impact operating conditions by affecting program and read bias. As illustrated in Figure 7d,i, due to the varying composition ratios of silicon nitride materials, a procedure exists to remove

all previously utilized interlayer silicon nitride material and subsequently fill the WL with tungsten material. Therefore, upon completion of manufacturing all final devices, it is anticipated that there will be no change in program and read bias attributable to silicon nitride materials of differing composition ratios, as these silicon nitride materials will be entirely absent. Consequently, although there may be differences in program bias caused by modifications to the cell structure due to silicon nitride with varying composition ratios, the operating conditions will remain unaffected by the material composition. The program bias required to achieve the same programmed Vth in the rectangle structure compared to the actual chamfer 2 was found to be insignificant, with a difference of approximately 0.2 V. This approach suggests that by controlling the physical properties of nitride during deposition, it is possible to manufacture 3D NAND flash with a confined nitride trap layer in a rectangular structure, minimizing Z-interference.

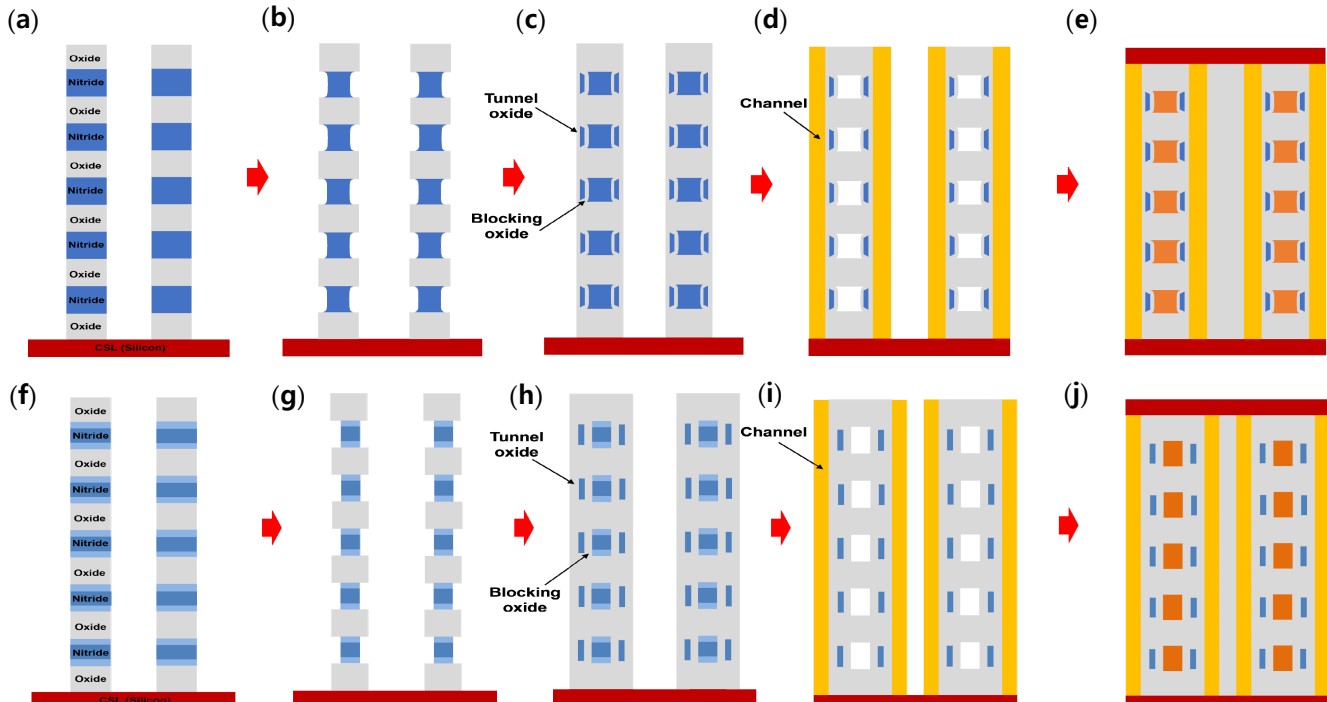

**Figure 7.** Typical major process flow to fabricate the confined nitride trapping layer structure in 3D NAND flash. (**a**) Oxide and nitride layers deposition. (**b**) A sidewall lateral etching of nitride. (**c**) Blocking oxide, SiN trapping layer, and tunneling oxide deposition. (**d**) Poly channel deposition and nitride wet etching. (**e**) Metal CVD, oxide, and silicon deposition. The devised major process flow for the confined nitride rectangle structure in 3D NAND flash. (**f**) Oxide and two nitride layers with different selectivity deposition. (**g**) A sidewall lateral etching of nitride. (**h**) Blocking oxide, SiN trapping layer, and tunneling oxide deposition. (**i**) Poly channel deposition and nitride wet etching. (**j**) Metal CVD, oxide, and silicon deposition.

## 5. Conclusions

This paper investigates Z-interference based on the confined nitride trap layer structure in 3D NAND flash. In 3D NAND flash, the Z-interference results vary significantly depending on the confined nitride trap layer structure. Notably, the rectangle structure, featuring a shorter distance between trapped electron charges of adjacent cells, exhibited superior Z-interference compared to the chamfer structure, characterized by a longer distance between the trapped electron charges of adjacent cells. The disparity is attributed to the fact that the increase in the effective gate length of the victim cell in the rectangular structure more than compensates for the deterioration caused by the attack cell. Furthermore, a proposed process flow aims to ultimately enhance Z-interference in the confined trap layer structure.

**Author Contributions:** Conceptualization, J.K.P.; methodology, J.K.P.; software, Y.K.; validation, S.K.H.; formal analysis, Y.K. and S.K.H.; investigation, Y.K.; writing—original draft preparation, Y.K.; writing—review and editing, J.K.P.; visualization, Y.K.; supervision, J.K.P.; project administration, J.K.P.; funding acquisition, J.K.P. All authors have read and agreed to the published version of the manuscript.

**Funding:** This study was supported by the Research Program funded by the SeoulTech (Seoul National University of Science and Technology).

**Data Availability Statement:** Data is contained within the article.

**Conflicts of Interest:** The funders had no role in the design of the study; in the collection, analyses, or interpretation of data; in the writing of the manuscript; or in the decision to publish the results.

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
