# Peer review of "Optimizing Confined Nitride Trap Layers for Improved Z-Interference in 3D NAND Flash Memory"

_electronics, doi:10.3390/electronics13061020_

Round 1
Reviewer 1 Report
Comments and Suggestions for Authors
This an good innovative paper. A proposed is given in terms of the shape of the SiN/SiO2 layer. Readers would be interested in the results and the related simulation profile, especially the effect on electronics distribution due to the shape variation. I only have two minor comments.
1. Fig.3 shows chamfer 1 and 2 would generate similar Vt shift on victim. This variation is clearly smaller than the effects as victim in Fig. 5. What if I argue that this variation is due to the area value, rather than the shape. Because chmfer 1 and 2 have similar area and both are smaller than the rectangle.
2. The author proposed a process innovation to use alternating oxide. Sicne you used more SiN, is there a need for large bias voltage, such as Vpgm, or Vread?
Author Response
List of changes and answers to the comments from Reviewer #1.
Q1. Fig.3 shows chamfer 1 and 2 would generate similar Vt shift on victim. This variation is clearly smaller than the effects as victim in Fig. 5. What if I argue that this variation is due to the area value, rather than the shape. Because chmfer 1 and 2 have similar area and both are smaller than the rectangle.
-> Thank you for your valuable comments, which are beneficial for further analysis. While conducting the experiment, Chambers 1 and 2 had distinct shapes, yet their areas were set to be completely the same. Thus, we found it challenging to account for all the minor variations in Fig. 3 solely based on area. Instead, as depicted in Fig. 4(b), it was determined that the impact of electron trapped charge distribution on the attack cell held greater significance.
Q2. The author proposed a process innovation to use alternating oxide. Sicne you used more SiN, is there a need for large bias voltage, such as Vpgm, or Vread?
-> In the proposed process, SiN materials with varying composition ratios were employed. However, as depicted in Fig. 7(i) and Fig. 7(d), there exists a procedure to eliminate all previously utilized interlayer SiN material to subsequently fill the WL with tungsten material. Therefore, after all final devices are manufactured, it is expected that there will be no change in Vpgm or Vread due to SiN materials of these different composition ratios because there will be no SiN materials at all.

Reviewer 2 Report
Comments and Suggestions for Authors
1. CTN is short for ? Hope the author can add notes on CTN.
2. how does the program happened? just 1 pulse program the attack cell to target VT? or it uses pulses by pulses?
3. inside figure 3d, between chamfer 1 and 2. below 1.5V, chamfer1 has smaller delta Vt, above 1.5V, chamfer2 has smaller delta Vt. How to explain such phenomenon.
4. with modified ONON layers, in figure 7f, the expectation is that it changes the etching rate so as to form rectangular structure. but there is no evidence provide here to prove the idea. Could the author add some references in this portion or adding some experimental data?
Author Response
List of changes and answers to the comments from Reviewer #2.
Q1. CTN is short for ? Hope the author can add notes on CTN.
-> Thank you for your feedback. We appreciate the reviewer's comments and have incorporated them by providing an explanation of CTN (charge trap nitride) in the revised manuscript.
Q2. how does the program happened? just 1 pulse program the attack cell to target VT? or it uses pulses by pulses?
-> At first, the programmed Vth of the attack cell was confirmed through ISPP operation. Afterwards, for the convenience of simulation, 1 pulse was applied to proceed with the program.
Q3. inside figure 3d, between chamfer 1 and 2. below 1.5V, chamfer1 has smaller delta Vt, above 1.5V, chamfer2 has smaller delta Vt. How to explain such phenomenon.
-> In Fig. 3d, the phenomenon where the Vth of Chamfer1 changes more significantly than that of Chamfer2 above 1.5V has been previously described in the text. Specifically, when Chamfer1 is located in the attack cell, the electric field originating from the attack cell becomes more concentrated at the bottom of the attack cell during a read operation. Consequently, because this concentration reduces the electron concentration in the space region more than Chamfer2, it is inferred that the degradation in Chamfer1 is greater when more electrons are stored above 1.5V. Please refer to the manuscript below for further details.
->“However, in the chamfer1 structure, the electric field of the attack cell is rather concentrated at the bottom of the attack cell, causing a significant drop in electron concentration in the space region. Therefore, this means that a larger voltage must be applied to the victim cell during a read operation to allow the same cell current to flow, and it can be assumed that the Z-interference phenomenon in chamfer1 is deteriorated compared to chamfer2 due to this cause. Due to this phenomenon, upon examining Figure 3(d), it is evident that there is no significant difference in Z-interference between Chambers 1 and 2 until the Program Vt of the attack cell reaches approximately 1V. However, it can be inferred that as the program level exceeds 1V, the difference in Z-interference be-comes more pronounced due to the dispersion effect of the electric field caused by the electrons in the charge trap layer.”
Q4. with modified ONON layers, in figure 7f, the expectation is that it changes the etching rate so as to form rectangular structure. but there is no evidence provide here to prove the idea. Could the author add some references in this portion or adding some experimental data?
-> Thank you for your feedback. We appreciate the reviewer's comments and have incorporated them by providing some references in the revised manuscript.
â– In the original manuscript : “Notably, the nitride in the area adjacent to the oxide deposits a material with a higher etch rate than the nitride in the middle area. Subsequent sidewall lateral etching of the two nitride layers controls the slow etching problem of the nitride at the interface with the oxide, as seen in Figure 7(g). “
â– In the revised manuscript: (On page 9) “Notably, the nitride in the area adjacent to the oxide deposits a material with a higher etch rate than the nitride in the middle area [1-2]. Subsequent sidewall lateral etching of the two nitride layers controls the slow etching problem of the nitride at the interface with the oxide, as seen in Figure 7(g).”
- LONGJUAN, T., YINFANG, Z., JINLING, Y., YAN, L., WEI, Z., JING, X., YUNFEI, L. & FUHUA, Y. Dependence of wet etch rate on deposition, annealing conditions and etchants for PECVD silicon nitride film. Journal of Semiconductors, 30.
- PROVINE, J., SCHINDLER, P., KIM, Y., WALCH, S. P., KIM, H. J., KIM, K.-H. & PRINZ, F. B. Correlation of film density and wet etch rate in hydrofluoric acid of plasma enhanced atomic layer deposited silicon nitride. AIP Advances, 6.

Reviewer 3 Report
Comments and Suggestions for Authors
Author Response
List of changes and answers to the comments from Reviewer #3.
Q1. This study presents a thorough analysis and simulation of 3DNAND memory with improved nitride layer design. Overall, the issue is valuable and intesreting, and the contents is well-organiozed. Thus, I recommend the publication if some small concenrs can be addressed. About the process of the SiN trap layer in Fig. 7(c) and 7(h), the reviewer is suspicious about the details of the formming of this layer. Maybe, some more infrmamtions should be given in the maunscioprt espcailly for the readers who are not familar with the 3D NAND process flow.
Basically, after the selective etching to nitride layers (e.g., in Fig. 7b), a blocking oxide fully-cover the deep-hole. then SiN trap layer cover-over the blocking oxide. However, the uniform-covered SiN prepeared by CVD should also exist over the Oxide zone (i.e., besides the desierd nitride trapping layer red-dot markered in the following).
So how to remove the nitride outside the desired zone? Could you please give more explanations? or there are totally different process to achieve the structure shown in Fig. 7c?
-> Following Fig. 7b, by depositing blocking oxide and SiN, then proceeding with the SiN pullback process, nitride can be selectively retained only in the desired area. We appreciate the reviewer's comments and have included a more comprehensive explanation and reference for the SiN pullback process as follows.
â– In the original manuscript : “In this structure, blocking oxide and silicon nitride trapping layers are deposited, as depicted in Figure 7(c). The subsequent charge trap layer pullback process separates the trapping layer in the Z direction, followed by the deposition of tunneling oxide. “
â– In the revised manuscript: (On page 8) “In this structure, blocking oxide and silicon nitride trapping layers are deposited, as depicted in Figure 7(c). At this stage, silicon nitride trapping layer is deposited as thick as 20 nm. In this way, the nitride film deposited inside the plug hole fills the entire recessed region and flattens the entire inner surface of the plug. Therefore, by repeatedly performing dry etching and wet etching process, a confined trapping layer can be formed in a self-aligned manner.[1] This charge trap layer pullback process separates the trapping layer in the Z direction, followed by the deposition of tunneling oxide. ”
- FU, C.-H., LUE, H.-T., HSU, T.-H., CHEN, W.-C., LEE, G.-R., CHIU, C.-J., WANG, K.-C. & LU, C.-Y. A Novel Confined Nitride-Trapping Layer Device for 3-D NAND Flash With Robust Retention Performances. IEEE Transactions on Electron Devices, 67, 989-994.

Reviewer 4 Report
Comments and Suggestions for Authors
This paper investigates the Z-interference effect in 3D NAND flash devices by conducting TCAD simulation for three confined nitride trap layer structures. The paper clearly illustrates the motivation for investigating Z-interference in 3D NAND flash, and the findings lead to a new process flow for fabricate the confined nitride trapping layer structure in 3D NAND Flash.
Some comments are as follows:
1. The paper does not justify the usage of the three selected confined nitride trip layer structures. It would be better to cite recent papers related to these structures.
2. The paper does not discuss the overhead comparison between three structures. Do these structures incur similar overhead so that the manufactory them are similar?
3. How is the Z-interference in the large-scale memory (e.g., 1k by 1k memory array) in the real system?